# Effects of a Standing Program for Ambulatory Children with Myelomeningocele: A Single-Subject Design

**DOI:** 10.3390/healthcare13192545

**Published:** 2025-10-09

**Authors:** Marianne Hanover, Elizabeth M. Ardolino, Megan B. Flores

**Affiliations:** 1Doctor of Physical Therapy Program, University of St. Augustine for Health Sciences, San Marcos, CA 92069, USA; 2Department of Physical Therapy, Baylor University, Waco, TX 76706, USAmegan_flores@baylor.edu (M.B.F.)

**Keywords:** spina bifida, stander, home program, intervention, physical therapy

## Abstract

**Background/Objectives:** Children with myelomeningocele (MMC) often experience lower extremity muscular contractures, which can impact their functional mobility. While standing programs have demonstrated benefits for children with other neuromuscular conditions, there is limited evidence on their use in ambulatory children with MMC who have joint deformities. This single-subject design study examined the impact of a home-based standing program on two ambulatory children with MMC, focusing on lower extremity muscle flexibility, functional movement quality, gait velocity, and participation in daily activities. **Methods:** Two children participated in a multi-phase single-subject (ABABA) withdrawal design beginning with the baseline phase and then alternating between the intervention and withdrawal phases. The intervention consisted of 60-minute standing sessions, five days a week, using a sit-to-stand stander (STSS) with support and supervision from a physical therapist (PT) and the parent. Primary outcomes included goniometric passive range of motion (PROM) and 10-Meter Walk Test (10 MWT). Secondary outcomes included the Pediatric Neuromuscular Recovery Scale (Peds NRS) and the Pediatric Evaluation of Disability Inventory Computer Adaptive Test (PEDI-CAT). **Results:** Improvements in hip and knee muscle flexibility were observed during the intervention phases, with some loss during the withdrawal phase. Functional movement quality improved in both children. Gait velocity and participation in daily activity scores remained stable during intervention phases. Parental feedback reflected increased independence and high engagement with the home program. One child discontinued due to a heel injury, highlighting the need for individualized support. **Conclusions:** Personalized standing programs may improve muscle flexibility and functional movement quality in ambulatory children with MMC. Further research is warranted to determine the optimal dosing regimen, ensure safety, and assess long-term functional outcomes.

## 1. Introduction

Spina bifida results from incomplete closure of the neural tube during embryonic development and affects approximately 3.5 children per 10,000 live births in the United States [1]. The most common form, myelomeningocele (MMC), involves herniation of the spinal cord through the spinal canal, leading to neurological damage at, below, or above the lesion level [1]. Children with MMC often experience a range of impairments, including bowel and bladder dysfunction, pain, sensory deficits, reduced coordination, muscle weakness, paralysis, and orthopedic complications [1,2]. These impairments frequently result in activity limitations, such as difficulty standing or walking independently. Secondary complications, including joint instability and muscular contractures in the hips, knees, and ankles, may also develop [3].

Children with MMC who do not present with lower extremity contractures tend to exhibit greater independence in self-care, with the absence of contractures serving as a significant predictor of independent walking mobility [4,5]. Hip and knee range of motion (ROM) has been shown to predict walking activity in children with MMC [3]. Mobility tends to decline from early childhood into adolescence [6], and the use of a wheelchair full-time is associated with a reduced quality of life [7]. Enhancing lower extremity flexibility and ROM can improve functional activity [8].

Surgical release of muscle contractures in the knee joint has shown some benefit in improving walking ability [9]. However, ambulatory status is inversely associated with a history of hip or knee contracture releases [5], and recurrence of knee flexor contractures is common in children who do not ambulate independently [10]. Surgical interventions are also linked to reduced odds of independent ambulation [11] and carry risks such as postoperative infection, pressure sores, pathological fractures due to immobilization, and osteopenia [12]. Serial casting offers a non-surgical alternative, improving knee contractures by approximately 30° in children with MMC [8,13]. This method involves applying a long leg cast, which is removed and reapplied every 2–3 days over three weeks, gradually extending the knee. While effective, casting limits mobility and may cause pain, skin irritation, pressure sores, muscle atrophy, and bathing difficulties [14]. Functional gait training typically begins after casting, with long-term maintenance requiring the use of knee–ankle–foot orthoses [8]. The effects of prolonged positioning on hamstring extensibility have been studied in children with cerebral palsy using a knee orthosis for 30 min, 5 days per week, for 8 weeks [15]. The intervention required participants to remain sedentary during its administration. While it improved hamstring extensibility and reduced spasticity, it did not result in significant changes in gross motor function. Reported complications included knee swelling and muscle cramps.

Çankaya and Gunel demonstrated that an eight-week standing program in nonambulatory children with MMC, without joint deformities, resulted in improvements in static and dynamic balance, as well as enhanced functional activities such as lying, rolling, and sitting [16]. However, to date, no published studies have examined the effects of a standing program in ambulatory children with MMC who present with joint deformities. Given the potential risks associated with surgical interventions [11], and the demonstrated effectiveness of prolonged passive stretching via serial casting and knee orthoses, a standing program that emphasizes sustained stretching may provide a non-invasive alternative for enhancing muscle flexibility, strength, and functional mobility without the associated risks and limitations associated with surgical, serial casting, and orthotic use.

Most evidence supporting standing programs comes from studies involving children with cerebral palsy, muscular dystrophy, arthrogryposis, and spinal muscular atrophy [17,18]. Use of static or dynamic standing frames has demonstrated benefits for lower extremity flexibility, including improved hamstring ROM [19], hip ROM [20], and prevention of knee flexion contractures [21,22]. Parents have reported that supported standing prepared children for acquiring head and trunk control, as well as standing and walking [23]. Although evidence suggests that ROM can be improved through standing programs, to date, no studies have investigated whether functional gains accompany these ROM improvements.

Except for the study by Çankaya and Gunel [16], the impact of standing programs on muscle flexibility, functional movement quality, gait speed, and participation in functional activities in children with MMC remains unclear. This single-subject design study aims to evaluate the effects of a home-based standing program on ambulatory children with MMC who exhibit significant hip and knee ROM limitations. Specifically, the study investigates the impact of a standing program on lower extremity muscle flexibility, functional movement quality, gait velocity, and participation in daily activities in children with MMC.

## 2. Materials and Methods

The study was approved by the Institutional Review Board. Participants were eligible for inclusion if they had a confirmed diagnosis of the MMC form of spina bifida, were between 5 and 12 years of age at the time of enrollment, were able to walk short distances, and presented with limited ROM in hip and/or knee extension. Individuals were excluded if they had medical conditions unrelated to MMC that impaired their ability to stand or if they had restrictions that contraindicated participation in a standing program. These restrictions included, but were not limited to, bone fractures, severe osteoporosis, and significant cardiovascular or respiratory compromise. The modified Hoffer scale assesses the child’s ambulatory status using five categories [24,25,26], with the lowest score (level 1) indicating the highest level of functional walking ability [24]. Children with a modified Hoffer level of 5, indicating nonambulatory status, were excluded from the study. All parents signed a consent form, and children provided verbal assent.

### 2.1. Study Design

The study employed a multi-phase single-subject (ABABA) withdrawal design beginning with the baseline phase and then alternating between the intervention and withdrawal phases. A single-case experimental design was selected due to the small and heterogeneous nature of the participant population, as well as for ethical and feasibility constraints associated with evaluating a rehabilitation intervention intended for real-world, home-based application. This design allowed for the tailoring of interventions to meet the unique needs of each participant [27]. The ABABA design was chosen to strengthen internal validity by the additional cycle of intervention and withdrawal, to rule out maturation, and provide insight into the treatment’s durability [27]. Each participant was provided with a summary of their study results that they could discuss with their medical provider should they decide to pursue the purchase of a sit-to-stand stander (STSS, Altimate Medical. Inc., Morton, MN, USA).

### 2.2. Participant Profiles

Two Hispanic male participants with MMC participated in the study. Child 1 was an 8-year-old boy classified as Modified Hoffer Level 4, indicating therapeutic ambulation only. He was able to ambulate with bilateral knee–ankle–foot orthoses and a reverse walker for short distances indoors; however, he primarily relied on a manual wheelchair for mobility within his home and the community. Child 1 had no history of surgery or pain and received outpatient physical therapy services once per month. Hypotonicity, operationally defined as reduced resistance to passive movement, was noted in bilateral lower leg musculature. He stated his physical therapy goal was to “get stronger and walk more.” He enjoyed playing baseball, football, and video games.

Child 2 was an 11-year-old boy classified as Modified Hoffer Level 3, indicating household ambulation. He used bilateral floor reaction solid ankle–foot orthoses, with a carbon fiber posterior strut, and bilateral forearm crutches for indoor functional mobility and occasional outdoor activities. His primary mode of transportation in the community was via a manual wheelchair. Child 2 had a history of bilateral hip surgery and received outpatient physical therapy services twice weekly. Hypertonicity, operationally defined as having higher resistance to passive movement, was noted in bilateral hip and knee musculature; however, he had hypotonicity in both legs below the knees. His physical therapy goals were to “walk longer distances, not be as tired, and not need crutches or braces.” His hobbies included wheelchair sports such as basketball, sled hockey, and tennis. He also enjoyed music, video games, and cooking. Child 2 complained of bilateral hip pain that had progressively increased over time to a severe rating of 9/10 on the numeric rating scale (NRS) for pain [2], limiting his ability to walk. Shortly before starting the study, he was evaluated by an orthopedic surgeon who recommended additional hip surgeries. After further consultation, the orthopedic surgeon, physiatrist, and parents agreed it was reasonable to attempt a standing program first.

### 2.3. Materials

The study was conducted in each child’s home. A multi-position STSS was used for this study as it accommodates asymmetric lower extremity ROM and supports individualized pressure distribution in each leg. The hip, knee, and ankle positions in this stander can be adjusted to independently maximize the child’s full available range of motion at these joints. The STSS enabled multiple short bouts of loading and unloading to be easily achieved without compromising leg alignment. The STSS reduced the need for lifting the child, increasing the child’s independence with transfers into and out of the device, which investigators felt would support long-term compliance with the home program. The STSS was delivered to each home and used exclusively during the B1 and B2 intervention phases. All study phases, baseline (A1), intervention (B1 & B2), and withdrawal (A2 & A3), were supervised. The same PT conducted study measures throughout all phases of the study and reassessed the child’s position in the stander during biweekly visits to the home. This assessment incorporated recommendations from both clinical evidence in the literature [19,21,28] and author recommendations provided in a systematic review of pediatric supported standing programs [17]. The initial assessment and subsequent biweekly reassessments included ensuring biomechanical alignment with adjustments as needed at the knees, ankles, and feet, to ensure full and equal loading of both legs and avoid direct pressure on the patella and tibial tubercle. The parent was provided with instructions and practiced assisting the child into the STSS position before the start of the intervention phases. They were responsible for supporting and guiding the child into this recommended position daily during the intervention phases of the home-based standing program between PT visits. Child 1 utilized a manual STSS, which required the parent to assist the child into a standing position. Child 2 used a STSS with a power lift option, allowing either the child or the parent to adjust the device to achieve standing.

### 2.4. Outcome Measures

Standardized materials were used for all outcome assessments across all study phases. These included a goniometer, an adjustable bench, a plastic cup, coins, and a rope to mark walking distances. Primary outcomes included passive range of motion (PROM) as a measure of muscle flexibility and the 10-Meter Walk Test (10 MWT) as a measure of gait velocity. PROM of hip extension and knee extension was measured in supine on a raised mat by the PT using a universal goniometer. The Thomas test was used as a measure of hip flexor length. The child was assisted into a supine position by the examiner with one knee flexed. Flexion of the contralateral knee was adjusted until a neutral lumbopelvic tilt was achieved. The parent maintained this position while the PT stretched the contralateral hip to the maximum available range. The stationary arm of the goniometer was placed parallel to the long axis of the trunk, and the movable arm was placed parallel to the long axis of the femur of the test leg [29]. The popliteal angle was used as a measure of knee flexor length. In supine, the hip of the test leg was flexed to 90 degrees and the knee extended to the maximum available range while the parent maintained the contralateral leg in extension. The stationary arm of the goniometer was placed parallel to the long axis of the femur of the test leg, and the movable arm was placed parallel to the long axis of the fibula [29].

PROM of each leg was measured biweekly before the PT assessed the child’s position in the STSS and made adjustments to the setup of the STSS to accommodate any changes in flexibility. Goniometric assessments have demonstrated good reliability and validity in school-aged children [30,31] and in children with cerebral palsy, with measurement errors ranging from 10 to 15 degrees [31]. However, a more specific value of 7 degrees has been reported as indicating true change for the popliteal angle [31].

The 10 MWT was utilized to measure gait velocity as it is a valid and reliable assessment in children with neurological conditions who use assistive devices and orthoses [32,33]. The minimal clinically important difference (MCID) for the 10 MWT in neurological cases varies depending on the specific condition and population being studied. A study on adult patients with an incomplete spinal cord injury determined that a change of 0.15 m/s was considered clinically significant [34].

Secondary outcomes included the Pediatric Neuromuscular Recovery Scale (Peds NRS) and the Pediatric Evaluation of Disability Inventory Computer Adaptive Test (PEDI-CAT). The Peds NRS is designed to evaluate the quality of functional movement in sitting, upper extremity use, standing, and walking across nine items on a 12-phase scale for children with MMC aged 1–12 [35,36]. Reliability and validity have been established for children with MMC [35]. The Peds NRS was administered in person by the PT, with scoring oversight via video by a trained evaluator to ensure consistency. The PEDI-CAT, a questionnaire completed online by parents, was used to assess two domains of everyday function: Daily Activities and Mobility. This scale is valid for children with spinal impairments [37,38]. The MCID has not been established for the Peds NRS or the PEDI-CAT; however, the minimal detectable change has been determined for the PEDI-CAT in children with neurological diagnoses in an acute care setting as 2.5 for mobility and 2.2 for daily activities [39].

### 2.5. Procedures

To support internal validity, all outcome measures were administered using the same tools and standardized setup within the home environment of each participant, and both children were assessed by the same PT. Both participants completed the 10 MWT at their preferred speed in the same home location using their respective assistive devices to ensure consistency in measurement. Before every session, checklists were reviewed to ensure protocol fidelity and consistency in data collection. Parents and participants were encouraged to maintain their regular daily routines.

During the intervention phases (B1 & B2), a structured standing home program was introduced. The target dose was 60 min of standing, five days per week, which could be completed in shorter, multiple sessions throughout the day. The recommended dosage was based on a systematic review by Paleg et al. that recommended daily standing sessions of 45 to 60 min to enhance hip, knee, and ankle ROM across various pediatric diagnoses [17]. This dosage also exceeded or aligned with most studies, regardless of their purpose, except for those aimed at improving bone mineral density [40].

Parents were encouraged to select a time that best fit their family’s schedule and plan an activity while the child stood. Child 1 often played or watched baseball while Child 2 either helped his mom with dinner preparation or completed his homework. Both children wore orthoses and shoes while standing to maintain a neutral foot position. This setup was intended to ensure that stretching occurred at the knees and proximal musculature, rather than encouraging overpronation at the foot. Comprehensive instructions on device safety, correct positioning, and signs of adverse effects were provided to parents. If children exhibited signs of discomfort (e.g., facial grimacing or excessive trunk movement), the stander was lowered slightly to achieve a tolerable position. The parents and children were instructed to maintain this position for 10–15 min before gradually increasing the height within the same session or in subsequent sessions, using the same feedback-based method.

During the initial setup visit for B1, each child stood for 15 min, and the PT confirmed that there was no skin irritation or discoloration at the contact points, including under the orthoses. To promote safety and gradual adaptation, the PT recommended starting with shorter, more frequent sessions during the early weeks of the program. Throughout all phases, parents were instructed to maintain a written daily log, documenting the child’s therapeutic, recreational, and sports activities. The same PT conducted all of the assessments in both children’s homes. Primary outcome measures were assessed every two weeks from baseline to completion, and secondary outcome measures were administered every four weeks from baseline to completion, ensuring fidelity of the study protocol.

## 3. Results

The total standing time for Child 1 during intervention phases was 7058 min over 118 days, averaging 59.8 min per session. He took breaks every 20 min during standing bouts at the beginning of B1 and was eventually able to easily stand for 60 min without rest toward the end of B2. Eight standing sessions across both intervention phases were missed due to other activities (e.g., swimming, baseball) or illness.

Child 2’s total standing time was 3000 min for intervention session B1 over 50 days, averaging 60 min per session. He missed 6 days of standing due to other activities (e.g., basketball, family outings). Two weeks into the first intervention phase (B1), Child 2’s NRS for bilateral hip pain decreased to 0/10 on a numeric pain scale, which was maintained throughout the phase. During the withdrawal phase A2, four days after completing B1, Child 2 started experiencing pain in his back (6/10), knees (7/10), and thighs (7/10). Study investigators consulted with the family and agreed that it was in the best interest of the child to begin B2 to help relieve his pain, restarting the standing program per the aforementioned guidelines. Two days after initiating B2, Child 2 stood for 6o min in the stander without a break, and his mother noted some discoloration in the right medial heel but did not contact the PT or pause the standing program as instructed. After standing again for 60 min two days later, his mother contacted the PT because he developed a blister on his right medial heel that was painful to touch. The researchers decided to discontinue the standing program and advised the family to seek medical advice.

Table 1 shows the changes in lower extremity muscle flexibility based on goniometric PROM measurements for Child 1. For Child 1, hip extension PROM increased 12 degrees on left hip and 9 degrees on the right hip after the first intervention phase, B1, and showed an increase of 2 degrees on left hip and 15 degrees on right hip after the second intervention phase, B2. Child 1’s knee extension PROM increased 10 degrees on the left knee and decreased 7 degrees on the right knee after B1 and increased 4 degrees on the left knee and 2 degrees on the right knee after B2.

Table 2 shows PROM results for Child 2. Hip extension PROM measurements after B1 intervention phase showed an increase of 4 degrees on the left hip and 10 degrees on the right hip. Child 2’s knee extension PROM decreased by 1 degree on the left knee and increased by 15 degrees on the right knee after B1 intervention.

Measurements of gait velocity revealed very little change for Child 1 throughout the study. He showed a slight decrease in gait speed after intervention phase B1, from 0.188 m/s to 0.152 m/s, and a slight increase in gait speed after intervention phase B2, from 0.111 m/s to 0.163 m/s. Child 2 showed an improvement in gait speed during the intervention phase B1, increasing from 0.86 m/s to 0.98 m/s; however, this change did not meet the 0.15 m/s minimal clinically important difference. There was, however, a significant decrease in gait speed during the withdrawal phase A2. A visual representation of gait speed during all phases of the study is shown in Figure 1.

Secondary outcome measure results for both children revealed that movement quality, as measured by Peds NRS, showed minimal but general improvement throughout the study. Child 1’s Peds NRS increased slightly from 9.42 to 9.67 during baseline phase A1, with changes seen in sit-to-stand and static standing. These scores remained stable throughout intervention phase B1, then showed a small increase to 9.92 during withdrawal phase A2, indicating improvements in standing stability and walking. Scores reached and maintained a peak of 10 during intervention phase B2, with improvements in static standing kinematics. Child 2 maintained Peds NRS scores throughout his participation, ranging from 10 to 10.33, with changes observed in sit-to-stand, static standing, and walking. The secondary outcome measure scores for Peds NRS and PEDI-CAT are shown in Table 3.

PEDI-CAT scores were a representation of activity and participation throughout the study. Scores for Child 1 in the Daily Activities domain showed minimal variation, ranging from 51 to 56, with no clear upward or downward trend. Child 2’s Daily Activities scores increased slightly from 56 to 60 by Week 12. Child 1 demonstrated stable Mobility scores ranging from 51 to 55 throughout the study and both intervention phases, with the exception of a single outlier occurring during the final withdrawal phase, which could not be explained. Mobility scores for Child 2 remained consistent, ranging from 60 to 63 during study participation.

Anecdotally, parents were enthusiastic about incorporating the standing device into a home-based standing program. The father of Child 1 shared that after two weeks of standing during Phase B1, “I was surprised that he [Child 1] could stand holding onto support with both arms and was able to kick a ball without his braces (KAFOs) on”. At the same point in the study, Child 1 stated, “I can balance myself.” By the end of Phase A1, both parents observed increased muscle bulk in his legs, and Child 1 noted that he was “swinging better for baseball”.

The mother of Child 2 described the program as a “win-win” for managing hip pain. She explained that it did not interfere with family time and helped avoid surgery. She found it easier as a caregiver because she only needed to assist with the straps, after which her son could complete the program independently. She also observed a marked increase in his walking endurance during the intervention phase. Although she acknowledged that finding the 60 min recommended for standing could be challenging, she emphasized that incorporating games, technology, homework, and hobbies made the program more manageable.

Child 2 described the program as “a pain reliever,” adding, “It was rough initially, but helped in the long run to give a good stretch.” When asked how it differed from other home exercise programs, he stated, “I don’t have to wait for someone to help me stretch; it is always available.” He felt empowered to push himself beyond what he had achieved with traditional stretching and appreciated not having to worry about doing the stretch incorrectly. When asked if he could see himself using the device long term, he responded, “Yes, 100%.”

## 4. Discussion

There is a paucity of information regarding interventions to improve lower extremity ROM and quality of functional mobility in children with MMC. Standing programs have been shown to be effective in other neuromuscular disorders [17]. However, there is no evidence to support their use in the ambulatory MMC population with already developed joint deformity. Therefore, the purpose of this single-subject design was to investigate the impact of a home standing program on ambulatory children with MMC who have significant hip and knee ROM limitations, limiting their lower extremity muscular flexibility.

Small improvements in hip and knee flexor muscle PROM during intervention phases and partial regression during withdrawal periods for Child 1 support the potential efficacy of the intervention for enhancing lower extremity PROM. Child 2’s short-term PROM gains also indicate positive responsiveness, especially during the initial intervention. These findings are consistent with those reported by Paleg et al.’s systematic review, which showed that children with spastic cerebral palsy who participated in standing programs improved their hip extension, knee extension, and hip abduction ROM [17]. These findings are also consistent with those of Laessker-Alkema et al., who found significant increases in ROM of the hamstring muscles after an 8-week bilateral knee orthosis intervention [15]. Additionally, Townsend et al. noted increased hip and knee flexor muscle length in boys with Duchenne muscular dystrophy who engaged in a home standing program over eight months [41]. The amount of ROM in the lower extremities has been shown to be related to walking activity and pain in children with MMC. Lullo et al. found that hip flexor flexibility PROM was the most significant clinical factor related to walking ability in children with MMC [3]. Decreased knee extensor active ROM has been correlated with painful knees in young adults with MMC, which may impact ambulation [42].

Clinically significant improvements in gait speed were not observed in this study. Child 1, who had a lower baseline gait speed, showed a minimal improvement, suggesting possible limited responsiveness for lower-functioning children or perhaps a floor effect. The lack of change in gait velocity of the younger participant, Child 1, on the 10 MWT may have been influenced by a lack of motivation at times, which was also noted as an issue in a study by Graser et al. in younger children [32]. Child 2 demonstrated a clinically significant [3] rapid decrease in gait speed during the first two weeks of the withdrawal phase, which may correlate with the 10-degree loss of R knee extension ROM and recurrence of lower extremity pain. These findings are consistent with the literature, which indicates that decreased knee extension ROM has been correlated with painful knees in young adults with MMC, which may impact ambulation [42]. Findings from this single-subject design study suggest that the intervention may have had a positive short-term effect on gait speed in Child 2; however, the improvements were not sustained. There was, however, a rapid decrease in speed during the withdrawal phase that was clinically significant [34] and may correlate with his recurrence of lower extremity pain. Although Child 2 showed minimal improvement in 10 MWT during the intervention phase B1, his mother observed improved walking endurance during this time, which could indicate that different aspects of walking function changed during the intervention phase. Outcome measures, such as the 6-Minute Walk Test (6 MWT), might have better captured changes in gait capacity, specifically muscular endurance [34].

Another factor to consider when designing a home standing program is the quality of functional movement. Child 1 reported the need for frequent seated breaks during the 60 min sessions due to trunk fatigue. He was only able to complete a full standing session without rest during the second intervention phase (B2), which aligned with improvements in Peds NRS scores. These findings are similar to those of Çankaya and Gunel, who found improved trunk control after a short standing program in children with MMC [16]. Child 2, who had higher ambulatory function, as indicated by a lower Modified Hoffer level and higher scores on the Peds NRS, quickly achieved the recommended dosage of 60 min without seated rest breaks; however, he exited the study before completing B2 due to an adverse event. Whether extending the intervention for Child 2 would have led to improved quality of movement remains unclear. Based on our limited results, a home standing program of at least 60 min may lead to meaningful improvements in the quality of functional movement in children classified at Hoffer levels 3 or 4. This finding aligns with the results of Verschuren et al., who reported that the standing position elicits muscle activation levels comparable to light physical activity in children and adolescents with cerebral palsy who experience prolonged sedentary behavior [43].

These two cases highlight the complex and varied presentations among children with MMC, underscoring the need for individualized home standing program dosing based on each child’s impairments. For example, each child had a different presentation of muscle tone and a unique surgical history. In hindsight, Child 2 may have benefited from shorter initial bouts within each standing session and more frequent reminders about the importance of skin checks because of his hypertonicity. Additionally, changes such as rapid growth or the introduction of new orthoses should be treated as potential risk factors, as they may increase skin pressure and lead to adverse outcomes. Early identification of skin issues is critical for preventing complications. Given the participants’ need for seated breaks, muscular ROM restrictions, and ability to assist with transfers, use of a STSS was essential. Child 2 used a powered unit, allowing him to independently adjust the stander height, thus increasing his autonomy and comfort while encouraging within-session adjustments to increase ROM. However, this independence also introduced risks. In this case, both Child 2 and his parent were highly motivated to resume the standing program following the first intervention phase due to their positive experience with pain reduction during the initial intervention period. However, without close supervision, he may have ignored recommendations to rest or inadvertently exceeded his available joint range, possibly leading to the adverse event. These observations suggest that powered standing devices should be used with caution in children with sensory impairments and may require close PT supervision to minimize the risk of adverse events. Additionally, shorter standing bouts with more frequent seated breaks may be necessary. Both cases highlight the importance of educating parents and children of a suitable age about the goals and risks associated with each standing device. These individuals should be trained to make adjustments to the device and should have access to a qualified PT who can assist with problem-solving to reduce joint deformities while emphasizing safety. This recommendation aligns with the evidence for family-centered models of care by pediatric rehabilitation services for children with spinal cord issues [44] and the importance of tailoring the care of children with brain injuries to their unique circumstances [45].

### Limitations

This study has several limitations. As a single-subject design with only two participants, it is inherently limited in its ability to control for threats to internal validity, as external factors unrelated to the intervention may have influenced the outcomes. Additionally, the small sample size limits external validity, preventing generalization of the findings to the broader population of children with MMC. Due to the limited number of participants, traditional group-based statistical analyses were not appropriate; instead, the effectiveness of the intervention was assessed primarily through visual analysis of data trends and level changes across phases. Also, the PT who assessed the children was not blinded to the children’s participation in the intervention. Using a second PT as a rater, blinded to the various phases, may have improved the accuracy of the measurements. While the reliability of using videos to score the Peds NRS has been documented [35,46], the accuracy of the findings might have been improved if the expert rater for the Peds NRS had physically performed the assessment. Finally, the study measurements were performed in the child’s home environment, presenting unique challenges compared to a controlled clinical laboratory setting, including a lack of standardization, observer bias, and variability in data collection schedules.

Future research should aim to include a larger, more diverse sample of children with MMC and should also examine pain, walking endurance, and quality of life.

## 5. Conclusions

This study demonstrated the feasibility of incorporating a standing program into the daily routine of ambulatory children with MMC and joint deformities. The varied results suggest that standing programs should be individualized and closely supervised by an experienced clinician. Education of both parents and children is essential. Home-based programs benefit from the expertise of a PT to help fit a child and provide individualized recommendations to minimize potential risks. Whether the benefits of improved muscular flexibility and quality of functional movement can be maintained when the standing program stops remains unclear.

## Figures and Tables

**Figure 1 healthcare-13-02545-f001:**
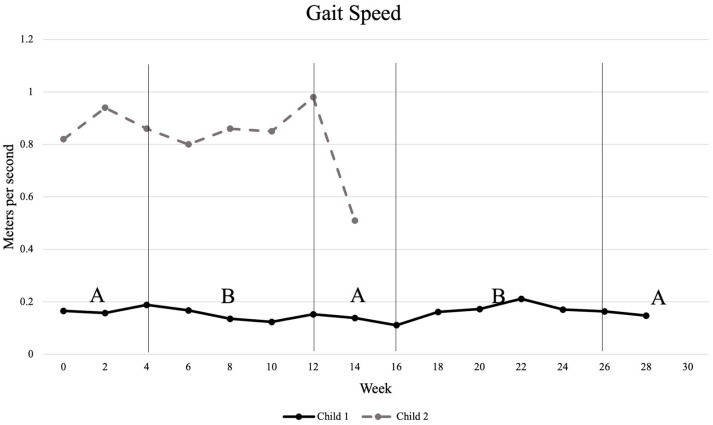
Changes in gait velocity as measured by 10 m walk test. “A” indicates phases with no intervention; “B” indicates intervention phases.

**Table 1 healthcare-13-02545-t001:** Child 1 passive range of motion results (degrees).

Phase	Week	Left Hip Extension	Right Hip Extension	Left Knee Extension	Right Knee Extension
A1	0	131	144	140	143
	2	145	152	143	156
B1	4	144	145	150	163
	6	144	154	153	160
	8	148	155	148	160
	10	158	163	152	158
	12	156	154	160	156
A2	14	155	163	139	150
B2	16	154	155	150	152
	18	151	158	153	160
	20	157	166	150	155
	22	161	163	155	154
	24	164	163	147	150
	26	156	170	154	154
A3	28	156	163	150	137
	30	147	150	155	153

**Table 2 healthcare-13-02545-t002:** Child 2 passive range of motion results (degrees).

Phase	Week	Left Hip Extension	Right Hip Extension	Left Knee Extension	Right Knee Extension
A1	0	159	156	142	125
	2	161	158	147	125
B1	4	164	158	141	120
	6	167	162	155	123
	8	166	163	144	125
	10	164	165	138	124
	12	168	168	140	135
A2	14	157	165	145	125

**Table 3 healthcare-13-02545-t003:** Secondary outcome measure results.

				Child 1	Child 2	Child 1	Child 2
		Child 1	Child 2	Pedi-CAT	Pedi-CAT	Pedi-CAT	Pedi-CAT
Phase	Week	Peds NRS	Peds NRS	Daily Activities	Daily Activities	Mobility	Mobility
A	Week 0	9.42	10.00	55	58	53	61
	Week 2	9.67	10.20	54	58	55	60
B	Week 4	9.67	10.20	54	56	51	61
	Week 8	NT	10.25	56	59	54	63
	Week 12	9.67	10.33	53	60	52	62
A	Week 14	9.67		54		53	
B	Week 16	9.92		53		52	
	Week 20	9.92		51		52	
	Week 24	10.00		53		50	
A	Week 26	10.00		54		41	

## Data Availability

The original contributions presented in this study are included in the article. Further inquiries can be directed to the corresponding author.

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
