# Peer review of "Effects of a Standing Program for Ambulatory Children with Myelomeningocele: A Single-Subject Design"

_healthcare, 2025, doi:10.3390/healthcare13192545_

Round 1

Reviewer 1 Report

Comments and Suggestions for Authors

Review – Major Revisions Required

This paper addresses an important clinical topic, as many children with myelomeningocele use standers and yet high-quality evidence is limited. However, the current manuscript requires significant revision before it could be considered for publication. At present, there are major methodological, reporting, and interpretive issues that must be addressed.

Study Design and Methods:
The rationale for selecting a sit-to-stand stander rather than a stander allowing positioning at full joint range is not explained. Please clarify whether passive range of motion (ROM) at each joint was measured prior to intervention, and how joint limitations influenced positioning. Without these data, it is unclear if participants were only stretched to the maximum of the least mobile joint. Additionally, the description of your design as ABABA or ABAB alternating baseline and withdrawal is inaccurate. A true single-case withdrawal design requires clearly defined baseline and intervention phases, which are not presented. Important details are missing: where standing occurred, who positioned the child, whether ROM was targeted daily or weekly, and how reliability was ensured between assessors.

Outcome Measures and Data Reporting:
Several outcome measures are not justified. For example, the Pediatric Neuromuscular Recovery Scale does not align with the stated goals of improving ROM. Exact ROM values for each hip and knee joint must be provided, both pre- and post-intervention, as well as during withdrawal phases. Reporting “slight gains” without numerical data or statistical analysis is inadequate. Similarly, the statement that standing reduced “discomfort” requires clarification and objective measurement. If one of two children discontinued due to heel injury, your study effectively has a 50% withdrawal rate and only one completer, which severely limits interpretability. Compliance data must also be presented (i.e., minutes of standing per day).

Presentation and Interpretation of Results:
Figures and graphs do not adequately present the data. ROM results must include minimum, maximum, and range values, with clear units. Gait speed changes should be contextualized relative to clinically meaningful differences and compared to normative values. Reporting changes of 2–4° is within measurement error and should not be described as meaningful without stronger justification. The discussion overstates the significance of findings and does not adequately address the limitations of sample size, design, and lack of statistical analysis.

Literature and Referencing:
There are inaccuracies and missing references in your literature review. Statements about surgical interventions (lines 57–59) require appropriate citations. The claim that standing is passive (lines 318–319) is incorrect; prior work (e.g., Vershuen and Israeli-Mendovic) demonstrates that standing can constitute exercise and should be cited. The paper should also reference existing literature on ROM improvement and comparable single-case studies in cerebral palsy for context.

Overall Assessment:
This paper has potential but in its current form reads more like a student project than a publishable study. Major revisions are required, including: (1) clarification of design and methodology, (2) full reporting of ROM and compliance data, (3) justification and alignment of outcome measures, (4) corrected statistical analysis with meaningful interpretation, (5) improved figures and data presentation, and (6) comprehensive revision of the literature review. Collaboration with a statistician is strongly recommended.

Author Response

Please see attached document for complete point by point response. 

Reviewer 2 Report

Comments and Suggestions for Authors

Abstract

In the results section, the results do not meet the proposed objectives. They should be revised to address all objectives. These results should be expressed numerically, but it is important to understand the gains that the programme has made in terms of the respective outcomes.

You should not repeat keywords that are already in the title.

Introduction

The introduction should be more substantiated and better justify the relevance of the study; this aspect should be revised.
In lines 68-70, contain important information, but have little impact and prominence in the introduction. I think it is important to review this part and give it greater prominence.

 Materials and Methods

I think you should explain very clearly why ABABA was chosen instead of ABAB and describe the  strategies used to reduce bias.

Since the study was conducted in each child's home, and parents were responsible for supporting 
and guiding the child during the standing programme at home during the intervention phases, how can this methodology not be biased? How do you justify this situation?

Won't the assessment carried out by parents be biased and limited? I think they should describe in even greater detail how this training was carried out to parents.

Results 

In this section, the text before Figure 1 and after table 1, is too long. I think it should be revised to make it more concise and summarised, including only data that is really important.

Could Figure 1 be more perceptible, as is Figure 2?

Table 2 is also unclear; it appears to be missing data. Decimal places should be standardised.

Review the results that claim to be clinically significant but lack consistent references in the text.

Discussion

They should pay attention to how they interpret data, particularly with regard to pain, and should be more careful and thorough.

I would like to see a more thorough discussion of the results of the walk, taking into account the results themselves. They should review this part. 

The adverse event must be analysed critically. Think about positive points and less positive points, but with benefits. Review this part, please.

Conclusion

In the conclusion, they should respond directly to the objectives.

Author Response

Please see attached document with line by line response to comments

Round 2

Reviewer 2 Report

Comments and Suggestions for Authors

Thank you once again for the opportunity to review the paper.
After the suggested changes and suggestions, the paper revealed greater potential in terms of content and scientific significance.
I now have greater clarity, with a common thread running throughout the work.
Congratulations, I enjoyed reading this version.